# Effectiveness and safety of nintedanib in prevention of pulmonary fibrosis induced by bleomycin in malignant ovarian germ cell tumour: study protocol for a randomised, double-blind, placebo-controlled trial

Sijian Li [1], Xinyue Zhang,[1] Min Yin,[1,2] Tianyu Zhang,[1] Rundong Zhang,[1] Jie Yang,[1] Jiaxin Yang[1]

For numbered affiliations see end of article.

**Correspondence to**
Jiaxin Yang;
yangjiaxin@pumch.cn

## ABSTRACT

**Introduction** Bleomycin is a crucial and irreplaceable chemotherapy regimen for malignant ovarian germ cell tumours (MOGCTs) but its toxicities especially pulmonary fibrosis have limited the dose of treatment efficacy and decreased the patients' quality of life (QoL). Nintedanib has been approved for treating progressive fibrosing interstitial lung diseases and has shown potential anti-tumour effects. This study aims to evaluate the effectiveness and safety of nintedanib in the prevention of pulmonary fibrosis induced by bleomycin in MOGCTs patients.

**Methods and analysis** This is a multicentre, randomised, double-blinded, placebo-controlled clinical trial. We will enrol a total of 128 patients who will be randomly assigned to the nintedanib group and placebo group in a 1:1 ratio. Standard bleomycin, etoposide and cisplatin chemotherapy will be given to each MOGCT patient. In addition, patients assigned to nintedanib and the control group will be given oral nintedanib 150 mg two times per day and placebo one tablet two times per day until 1 month after the last cycle of bleomycin therapy, respectively. The primary outcome is the decline of forced vital capacity (FVC). The secondary outcomes are the decline of other pulmonary function indices (forced expiratory volume in 1 s; FVC pred%, carbon monoxide diffusion capacity) and the patients' QoL, oncological and fertility outcomes. We will use electronic case report forms to record all the participants' data and SPSS V.27.0/STATA V.16.0/Graphpad Prism V.8.0 to conduct statistical analysis.

**Ethics and dissemination** The Ethics Committee of Peking Union Medical College Hospital has approved the study (I-23PJ400). Written informed consent will be obtained from all participants/guardians. Study results will be submitted to peer-reviewed medical journals for publication and presented at academic conferences.

**Trial registration number** ChiCTR2300070492.

## STRENGTHS AND LIMITATIONS OF THIS STUDY

⇒ This trial aims to evaluate the effectiveness and the safety of nintedanib in malignant ovarian germ cell tumours patients receiving bleomycin chemotherapy.

⇒ This prospective, randomised, double-blinded, placebo-controlled design increases the reliability of the results and conclusions.

⇒ The non-invasive test of pulmonary function and other outcomes will enhance the feasibility and compliance of participants.

⇒ The limitation is that it may need more time to enrol enough participants due to the rarity and need a long-term follow-up to assess some secondary outcomes and the single-centre nature.

ovarian malignant tumours that originate from the abnormal differentiation and malignant transformation of primordial germ cells of embryonic gonads. MOGCTs are rare and only account for about 3% of all ovarian malignancies, with an estimated incidence of 0.4/100 000 women years and mostly affected patients about 20 years.[1 2] It is estimated that approximately 60%–70% of MOGCTs patients are diagnosed with International Federation of Gynecology and Obstetrics (FIGO) stage I, and only less than 5% are with FIGO stage IV diseases on initial treatment, therefore, the prognosis of MOGCTs patients is usually satisfactory.[1 3]

Currently, the standard treatment for MOGCTs patients can be a combination of surgery and adjuvant chemotherapy (bleomycin, etoposide and cisplatin, BEP regimen) based on tumour stage and/or grade, while fertility-sparing surgery is always recommended in childbearing or young patients.[4 5]

## INTRODUCTION

Malignant ovarian germ cell tumours (MOGCTs) are heterogeneous non-epithelial

Bleomycin is the crucial and irreplaceable regimen for chemotherapy in MOGCTs.[6] However, the toxicities associated with bleomycin including acute and later-onset pulmonary toxicity have limited the dose of the regimen, in terms of treatment efficacy and have decreased the quality of life (QoL) in these young patients.[7] The risk of pulmonary fibrosis is an important adverse effect (AEs) of bleomycin, and it has been reported that pulmonary toxicity induced by bleomycin occurred in 6.8%–46% of patients, with bleomycin-related mortality of about 3%–5%.[8] Other AEs are less common, such as fever, hypersensitivity and organising pneumonitis.[9]

Tumour relapse usually occurs within 2 years after treatment in MOGCTs patients.[10] Although chemotherapy after relapse can achieve a complete remission rate of about 75%, about one-third of advanced MOGCTs patients may experience recurrence or chemotherapy resistance in a short period, resulting in poor survival outcomes.[7 11] Relapsed MOGCTs are usually not curable partly due to intolerance to bleomycin or not having enough therapeutic doses of bleomycin, whereas other chemotherapy regimens are not effective.[12]

Hence, it has always been emphasised to alleviate bleomycin-induced toxicities, especially pulmonary fibrosis. Although the exact mechanisms of pulmonary fibrosis caused by bleomycin remain uncertain, it has been proposed that inflammatory reaction and subsequent fibroplasia may play an important role in the pathophysiology of pulmonary fibrosis.[13 14] Nintedanib is a small-molecule tyrosine kinase inhibitor that targets platelet-derived growth factor receptor-α and β, fibroblast growth factor receptor-1–3 and vascular endothelial growth factor receptor-1–3.[15] Nintedanib has been approved for the treatment of idiopathic pulmonary fibrosis (IPF) in many countries, and several clinical trials have proven its effectiveness and safety in the treatment of IPF.[16 17] Recently, Gundogan et al also reported a 13-year-old lymphoma patient treated with a bleomycin-containing chemotherapy regimen who developed bleomycin-induced pneumonia that was successfully treated with nintedanib.[18] Moreover, nintedanib also showed antitumour effects in some reports.[19 20]

However, none of these studies provide evidence of nintedanib in the treatment of pulmonary fibrosis in MOGCTs. Therefore, a prospective, double-blinded, randomised control trial is designed to evaluate the effectiveness and safety of nintedanib in MOGCTs patients receiving bleomycin.

## Objective

The primary aim of this randomised, control trial is to determine whether nintedanib can alleviate the reduction decline of forced vital capacity (FVC) in MOGCTs patients who receive BEP chemotherapy. The secondary objective is to evaluate the decline of other pulmonary function indices (forced expiratory volume in 1 s, FEV1; FVC pred%, carbon monoxide diffusion capacity, DLCO), the safety of nintedanib treatment, oncological and fertility outcomes, and QoL in MOGCTs patients.

## METHODS AND ANALYSIS

### Study design

This is a prospective, randomised, double-blinded, placebo-controlled clinical trial that was registered at the Chinese Clinical Trial Registry on 14 April 2023 (ChiCTR2300070492). It has been approved by the Ethics Committee of Peking Union Medical College Hospital and will be conducted in accordance with the Declaration of Helsinki and strictly follow the latest Consolidated Standards of Reporting Trials recommendations,[21] Standard Protocol Items: Recommendations for Interventional Trials statement[22] (online supplemental table 1). For all patients who will participate in this study, written informed consent will be obtained before treatment.

A total of 128 participants diagnosed with MOGCTs treated with BEP chemotherapy will be enrolled in the trial. The total screening and enrolling period will last for about 2.5 years in estimation, and they will be followed up for at least 1 year (up to 5 years) after treatment. The enrolment is planned to start on 1 October 2023 and end on 1 April 2026. The participants will be randomised and assigned to either the intervention group or the placebo group in a 1:1 ratio using a central randomisation system. In addition to standard adjuvant chemotherapy, they will be given nintedanib (150 mg) or placebo in granule form twice a day before chemotherapy till 1 month after the last cycle of chemotherapy.

### Participants

The inclusion criteria are listed as follows:
1. Newly diagnosed MOGCTs patients who will be treated with BEP chemotherapy or bleomycin-based chemotherapy.
2. Age between 14 and 40 years.
3. Pulmonary function test can be completed.
4. FVC≥45% of normal predicted value and DLCO≥60% of normal predicted value.
5. Patients who agree and sign the written informed consent, and with good compliance.

Patients who meet the following criteria will be excluded:
1. Patients with pre-existing interstitial lung disease.
2. Patients with severe medical diseases, severely impaired liver function (transaminase increased to 2.5 times the upper limit of normal) and severely impaired renal function (creatinine clearance rate <30 mL/min).
3. Combined with other reproductive system malignant tumours.
4. Complicated with deep vein thrombosis, pulmonary embolism, myocardial infarction, cerebral infarction and other diseases.
5. Combined with other diseases requiring long-term hormone therapy.

6. Patients with chronic obstructive pulmonary disease or pulmonary hypertension that influence pulmonary function.
7. Patients who are considered not suitable to be enrolled by researchers.

## Recruitment of patients and data collection
### Participants inclusion
The recruit proclamation that includes a summarised description of the study will be released through webpages, in inpatient and outpatient areas. All the patients with MOGCTs (or their guardians in the case of juveniles) who visited our hospital will be evaluated whether they are suitable for entry into the clinical trial. The final pathology confirmation will be determined by two experienced pathologists at Peking Union Medical College Hospital. Then, for those potentially eligible MOGCTs patients, the study team will ask them or their guardians whether they are willing to participate in the study. Once written informed consent is obtained, members of the research team will collect the basic demographic information. Otherwise, the reasons why patients are excluded will also be recorded.

### Study group assignment
After the exclusion and inclusion process, we will perform the concealed study group assignment based on the central randomisation system. An independent statistician will generate a randomisation sequence for all the eligible MOGCTs patients. The patients will be randomly allocated to either nintedanib (experimental) or placebo (control) groups. For all the MOGCT patients enrolled in this trial, a standard treatment, namely comprehensive fertility-sparing surgical staging, or radical surgical staging plus BEP chemotherapy for 3–4 cycles, in accordance with the ESMO guidelines,[5] no matter whether they are assigned to which group. In addition to standard treatment, patients allocated to the nintedanib group will be given oral nintedanib 150 mg two times per day before the first cycle of BEP chemotherapy until 1 month after the last cycle of BEP chemotherapy. Accordingly, patients in the control group will be given an oral placebo one tablet per time, two times per day before the first cycle of BEP chemotherapy until 1 month after the last cycle of BEP chemotherapy. The nintedanib and placebo will be provided by the study team and the appearance will be the same, without identifiable labels or packaging. The patients will be requested to record the medication and return the packaging to researchers after treatment. The flow chart (figure 1) shows the simplified procedure of group allocation, interventions and outcomes assessment.

### Termination of therapy (nintedanib or placebo)
1. The planned end of treatment (till 1 month after the last cycle of BEP chemotherapy).
2. Termination of treatment due to severe AEs.
3. Withdrawal of written informed consent.

## Management in case of patients who could not complete pulmonary function test
In patients who may not be able to receive pulmonary function tests due to bleomycin-induced acute lung injury, arterial blood gases and high-resolution pulmonary CT will be performed to indirectly examine the pulmonary function. Meanwhile, it is also an adverse event that will be recorded in this trial. Pulmonary function tests will be performed after recovery or until those patients can complete this examination. For patients whose general condition has deteriorated due to MOGCT progression, if such patients had been enrolled in this trial, they would have recorded baseline and dynamic pulmonary function tests until they could not finish the pulmonary test. If they have not already finished the complete treatment for MOGCT and cannot continuously receive bleomycin-based chemotherapy, these adverse events will be recorded as positive events during the trial to evaluate the safety of nintedanib.

## Follow-up data collection
The disease-specific data, including tumour stage, pathological subtypes of MOGCTs, tumour markers' value, surgical options and chemotherapy response, etc, will further be recorded in detail. The patients need to routinely follow-up in outpatient before each chemotherapy cycle. The frequency of outpatient follow-up after finishing chemotherapy is set as follows: every 3 months within 3 years after treatment and every 6 months between 3 and 5 years after treatment. Additional follow-up will be provided in case of special conditions, such as severe chemotherapy side effects or tumour recurrence. A telephone follow-up will be a supplementary option for those patients who follow-up in local hospitals or miss the appointment day. Importantly, the patients' smoking status, whether they receive oxygen therapy, and/or chest radiation before or during the treatment will be recorded in detail due to these factors may have a synergised effect with bleomycin.

During each follow-up, the patients will need to report the oncology outcomes, pulmonary function, QoL, menstrual conditions and fertility outcomes for childbearing patients with the desire for fertility. The interval between bleomycin-containing chemotherapy and the onset of lung injury will be exactly recorded during the trial. The QoL assessment will be based on the general conditions and self-report questionnaires listed as follows.

The privacy of patients will be rigorously protected throughout the whole study process. Any statistical analysis or publications of data can only be presented in an anonymisation format.

## Validated questionnaires used in the trial
QoL will be mainly assessed by the Functional Assessment of Cancer Therapy Scale General (FACT-G).[23] Other questionnaires include the Insomnia Severity Index (ISI),[24] the International Physical Activity Questionnaire,[25] Peripheral Neurotoxicity Questionnaire (GOG-NTX).[26]

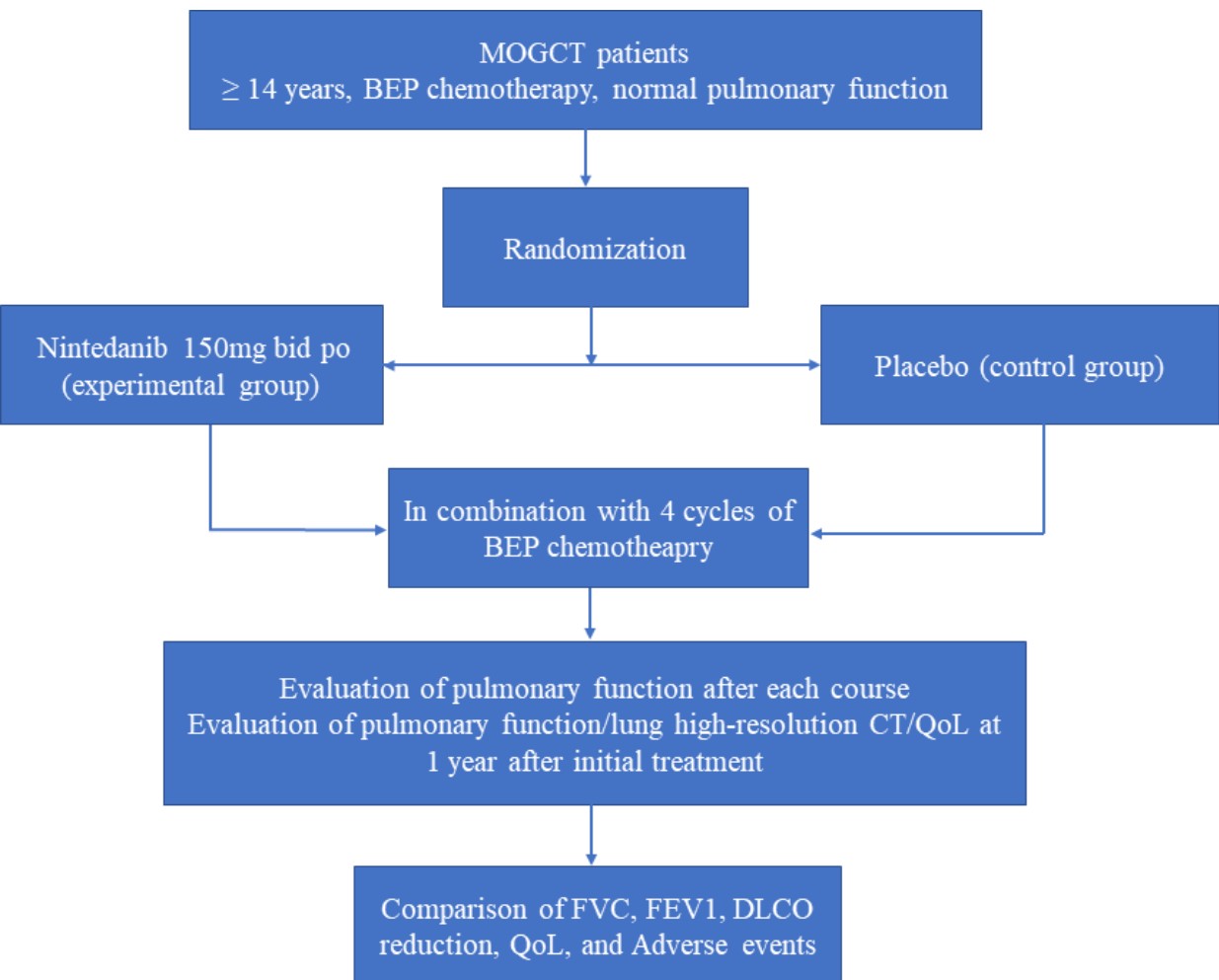

**Figure 1** The flow chart demonstrates the simplified process of group allocation, interventions and outcomes assessment. BEP, bleomycin, etoposide and cisplatin; DLCO, carbon monoxide diffusion capacity; EFV1, forced expiratory volume in 1 s; FVC, forced vital capacity; MOGCT, malignant ovarian germ cell tumours; QoL, quality of life.

The FACT-G is a symptomatic scale including 12 items that can evaluate patients' symptoms, chemotherapy AEs, sexual and hormonal situations, etc. Similarly, ISI includes seven items to assess the quality of sleep and the severity of insomnia. IPAG shows validation in measuring overall physical activity and its frequency and intensity. GOG-NTX uses 11 items to describe peripheral nerve disorders, such as numbness, discomfort, arthralgia, deep sensitivity problems and walking disorders. The detailed description has also been discussed in a previous study.[27] A combination of these questionnaires can help demonstrate the life status of MOGCT patients after treatment.

### Outcomes
#### Primary outcomes
The primary outcome of the study is the declined degree of FVC in MOGCT patients who receive BEP chemotherapy revealed by respiratory function test. The secondary outcomes are the decline of other pulmonary function indices (FEV1; FVC pred% and DLCO), the safety of nintedanib treatment, oncological and fertility outcomes, and QoL in MOGCTs patients.

The safety assessment of nintedanib is measured by patients' vital signs, auxiliary examinations and AEs. The auxiliary examinations include blood tests (routine analysis, liver and kidney function, etc), urine and stool examinations. We will record any AE during the treatment and follow-up period. The definition and classification of AEs based on the CTCAE5.0. Treatment will be continued in case of grade 1 or tolerable grade 2 AEs, and temporary termination of treatment will be applied in case of the first occurrence of intolerable grade 2 or grade 3 AEs until recovery to grade 0–1 or grade 2 AEs. However, permanent termination of nintedanib medication will be considered on the second time of intolerable grade 2 or grade 3 AEs, no matter the previous toxicities or newly identified toxicities.

### Sample size calculation
This study adopted a non-inferiority clinical trial design, and the sample size was calculated by using the sample size calculation website (https://sample-size.net). Based on the decline in lung function of patients treated with bleomycin at 1 year in the literature and the improvement

in pulmonary function with nintedanib,[17 28–30] we set the decline of FVC in the placebo group as over 8%. Based on the efficacy of nintedanib in the study of pulmonary fibrosis patients,[16] we set the decline of FVC in the nintedanib group as 4%. Using a non-inferiority cut-off value of 10%, a one-sided α value of 0.025, and the test power of 0.09, we calculate the sample size of a single group as 53. Considering the group set a 1:1 ratio in two groups and 10% of the dropout cases during the trial, at least 116 patients will need to be included (online supplemental figure 1). Moreover, we presume that no more than 10% of the enrolled MOGCT patients may be unable to continue to receive bleomycin-based chemotherapy due to the pulmonary fibrosis, we will recruit additional 10% patients (12 patients) to better conduct this trial. Therefore, we plan to include at least 128 patients in this trial.

### Patient and public involvement

We established a panel of eight patients (in case of adult patients) or their guardians (children or adolescents' patients) was established for this trial. They helped to modify the study design and improve the final study protocol during online focus group meetings. Particularly, they proposed the recommendations on patient-related outcomes including the minimally important clinical differences, methods of recruitment and the design of data collection. They will be actively involved throughout all critical stages of the trial through regular parent panel meetings. The results of the study will inform the participants or their guardians in case of children/adolescent's participants. Moreover, the additional burden of the intervention in this clinical trial will be fully covered by the hospital. However, the charge due to standard treatment of MOGCTs should be paid by the patients themselves.

### Data management and statistical analysis

We will use an electronic case report form to collect data, and an electronic database will be subsequently generated. All the investigators need to be authorised to access the data and will be required to maintain data confidentiality. A periodic indication to update outcomes' information during the follow-up will also be set up. After completion of the study, data will be stored for at least 20 years at a central data drive at the Peking Union Medical College Hospital. The datasets used and/or analysed during the current study can be obtained from the principal investigator on reasonable request. Moreover, a written, signed data-sharing agreement will be required to get data access.

The variables will be described according to the distributions, namely, mean±SD (range) for normally distributed continuous variables, otherwise they are described as medians and IQRs. Counts (percentages) will be used to present discrete variables. Methods for comparisons of variables between two groups will also be determined according to the data types. For oncology outcomes, including recurrence-free survival, overall survival and disease-specific survival, we will use the Kaplan-Meier method (log-rank test). The univariate analysis will be performed to screen factors that should be subjected to multivariate analysis, and we set the cut-off p values as 0.10 in univariate analysis. The Cox regression model or logistic regression model will be conducted in multivariate analysis. SPSS (V.20.0; SPSS), GraphPad Prism (V.8.0; GraphPad Software, La Jolla, California, USA) or STATA (SE V.12, StataCorp) software will be applied to perform the statistical analysis when appropriate. We set the statistical significance cut-off as 0.05 (two-tailed p value).

### Harms

The safety of nintedanib in treating IPF has been confirmed in previous clinical trials.[16] The AEs are usually mild and tolerable. The most common AEs of nintedanib therapy are as follows (incidence ≥5%): diarrhoea, anorexia, sickness, vomiting, abdominal pain, headache, elevated liver enzyme, weight loss or hypertension. Other rare AEs reported in previous research include haemorrhage, gastrointestinal perforation, elevated bilirubin levels and cardiac toxicity. However, currently, there has been limited data on the safety of nintedanib in MOGCT patients receiving bleomycin-based chemotherapy. Therefore, unpredicted AEs may be found during this trial, and all the AEs observed during the study period will be treated according to the clinical practice. Especially, for MOGCT patients receiving bleomycin-based chemotherapy, the benefit of prevention of bleomycin-induced pulmonary toxicities would be greater than the risk and the harms of nintedanib itself.

### Ethics and dissemination

The Ethics Committee of Peking Union Medical College Hospital has approved the study (I-23PJ400). Written informed consent will be obtained from all participants/guardians. Study results will be submitted to peer-reviewed medical journals for publication and presented at academic conferences. The study will be conducted according to the principles of the Declaration of Helsinki.

### Current status

Participants' enrollment is planned to start on 1 October 2023 and end on 1 April 2026.

## DISCUSSION

At present, bleomycin-induced pulmonary fibrosis remains a severe chemotherapy AE that significantly impairs the therapeutic efficacy and QoL in MOGCT patients. It was estimated that 8% of patients who received bleomycin may develop varied degrees of pulmonary fibrosis, and nearly 2% were lethal.[29] Although some optional chemotherapy regimens were proposed to decrease chemo toxicities in MOGCT patients, bleomycin is a crucial and irreplaceable regimen for chemotherapy in MOGCTs patients.[31 32] Nonetheless, the role of bleomycin is more important for recurrent MOGCTs due to dissatisfactory responses

to other regimens.[6 12] Therefore, how to alleviate pulmonary fibrosis during bleomycin-containing chemotherapy in MOGCTs has always raised great concerns.

The metabolism of bleomycin in the lung is poor because of the low activity of detoxifying enzymes and bleomycin hydrolase.[8] Hence, the accumulation of bleomycin in the lung, impairs the vascular system, resulting in the aggregation of inflammatory cells and fibroblasts into the lung tissue, which eventually leads to pulmonary fibrosis.[33] Nonetheless, the exact mechanisms of pulmonary fibrosis development remain unclear, and the time between bleomycin medication and pulmonary fibrosis varies greatly among different patients.[34] Routine prophylactic administration of antifibrosis drugs in patients receiving bleomycin may be practical due to the availability for early identification of high-risk patients. In several clinical trials of nintedanib in treating progressive fibrosing interstitial lung diseases or IPF, nintedanib showed a significantly lower reduction in pulmonary function.[18 35] Furthermore, nintedanib has been showing promising therapeutic efficacy in treating fibrosis caused by bleomycin-containing chemotherapy and potential additional benefits in combination with common chemotherapy regimens in cancer patients.[18–20] Besides, these studies showed relatively minor AEs of nintedanib, revealing the safety of medication in targeted treating patients.[35]

However, currently, there is limited evidence of nintedanib for the prevention or treatment of bleomycin-induced pulmonary fibrosis in MOGCT patients. There has been only limited low-grade evidence originating from case reports. Our prospective, double-blinded, randomised controlled trial aims to provide high-quality treatment on this issue, which will be helpful in the management of the severe AEs caused by bleomycin-containing chemotherapy in this population. Nevertheless, one major limitation of this study is the single-centre setting, further multicentre prospective trials should be conducted.

**Author affiliations**
[1]National Clinical Research Center for Obstetric and Gynecologic Diseases, Department of Obstetrics and Gynecology, Peking Union Medical College Hospital, Chinese Academy of Medical Sciences, Peking Union Medical College, Beijing, Beijing, China
[2]Department of Gynecology, Qilu Hospital (Qingdao), Cheeloo College of Medicine, Shandong University, Qingdao, Shandong, China

**Acknowledgements** We want to thank patient advisers and all the patients enrolled in this study.

**Contributors** JY is the principal investigator of this study who conceives the study design and modifies the manuscript. SL wrote the manuscript; TZ, XZ, MY and RZ completed the establishment of the eCRF, participated in the literature review and trial registration; JY participated in the study design and designed the statistical analysis methods. SL, TZ, XZ, MY and RZ will participate in patient recruitment, data collection and statistical analysis. All authors read and approved the manuscript.

**Funding** This study was supported by National High-Level Hospital Clinical Research Funding (2022-PUMCH-B-083) and Chinese Academy of Medical Sciences (CAMS) Innovation Fund for Medical Sciences (2022-I2M-C&T-B-023).

**Competing interests** None declared.

**Patient and public involvement** Patients and/or the public were involved in the design, or conduct, or reporting, or dissemination plans of this research. Refer to the Methods section for further details.

**Patient consent for publication** Not applicable.

**Provenance and peer review** Not commissioned; externally peer reviewed.

**ORCID iD**
Sijian Li http://orcid.org/0000-0003-1578-0516

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
