## [Reviewer comments · BMJ Open]

ARTICLE DETAILS

TITLE (PROVISIONAL)	Effectiveness and Safety of Nintedanib in Prevention of Pulmonary Fibrosis Induced by Bleomycin in Malignant Ovarian Germ Cell Tumor: Study Protocol for a Randomized, Double-blind, Placebo-controlled Trial
AUTHORS	Li, Sijian; Zhang, Xinyue; Yin, Min; Zhang, Tianyu; Zhang, Rundong; Yang, Jie; Yang, Jiabin

VERSION 1 – REVIEW

REVIEWER	Okamoto, Masaki Kurume University School of Medicine, Division of Respiriology, Neurology and Rheumatology, Department of Medicine
REVIEW RETURNED	06-Jun-2023

GENERAL COMMENTS	The authors showed the protocol of RCT for providing evidence of nintedanib in the treatment of pulmonary fibrosis in MOGCTs. This paper has been well written. The resolution of the following issues for acceptance of BMJ Open. Major comment 1. Please show the rationale for setting the suppression of decreased pulmonary function as the primary endpoint. For example, is there previous report that decreased pulmonary function over time among patients with malignant disease treated with bleomycin?2. In patients with bleomycin-induced acute lung injury, pulmonary function may not be measured due to severe symptoms. In addition, it may not be possible to measure pulmonary function in patients whose general condition has deteriorated due to progression of MOGCT. Please show how to deal with such reduction bias.3. Please explain the reason why time-to-event with the onset of lung injury as an event is not endpoint.4. Having chronic obstructive pulmonary disease or pulmonary hypertension influence pulmonary function. Patients with these disease were not included in the exclusion criteria.
--

REVIEWER	Zhang, Wenxi Third Military Medical University Southwest Hospital, gynecology and obstetrics
REVIEW RETURNED	11-Jun-2023

GENERAL COMMENTS	I think it's worthwhile trial, with complete design. I only have several suggestions.  1. The first one is about the age of the participants, which is 14-35y. Most of the similar trails set the age no older than 40 years old. As you mentioned in the limitations that it needs more time to enroll the enough participants due to it's rarity. I recommend extending the age of the included subjects to 40. 2. As I know the oxygen therapy, smoking, and chest irradiation are thought to synergize with bleomycin, I suggest the protocol point out that. 3. During the chemotherapy cycles, the participants may get pneumonitis and should stop the bleomycin. They can't finish the 3-4 cycles as the protocol mentioned, so how to deal with that? 4. As I know, the sample size calculation is due to the primary outcomes of the study. Since it has several primary outcomes, the calculation of each one should to be done to get largest sample size. The protocol only mentioned the FVC and DLCO, and they were calculated by the same rate. I read the references and did not find the rate for the DLCO. I suggest consulting statisticians for this part, or I would propose only one primary outcome for the study.
--

VERSION 1 – AUTHOR RESPONSE

Response to reviewer 1#: Thanks for your constructive comments, our responses are listed as follows:

Comment 1: Please show the rationale for setting the suppression of decreased pulmonary function as the primary endpoint. For example, is there a previous report that decreased pulmonary function over time among patients with malignant disease treated with bleomycin?

Response: One of the most devastating adverse effects of bleomycin in treating MOGCT is pulmonary toxicity, and the FVC is a very helpful and widely used index in pulmonary function tests. The following articles summarize the related data on bleomycin in reducing the decline in pulmonary function: PMID: 31566307; PMID: 26308607; PMID: 31285305; PMID: 27305276, etc. According to the results from three large RCTs (INPULSIS-1/2 and INBUILD), nintedanib showed a promising therapeutic effect in that it significantly reduced the decline in FVC in idiopathic pulmonary fibrosis (the declined FVC in the placebo group was about two times than in nintedanib group). In a systematic review reported by Delanoy et al., 7.7% of the 221 patients with ovarian sex cord-stromal tumors receiving bleomycin developed bleomycin-induced pneumonitis. This result was comparable to the data from 835 germ cell tumor patients receiving bleomycin. Roncolato et al. in 2016 also reported that nearly all patients had FVC/DLCO decline, and 10% of 43 patients with germ cell tumors receiving bleomycin had an FVC decline of over 15%. Moreover, we modify the primary outcome as the decline of forced vital capacity (FVC). The secondary outcomes are the decline of other pulmonary function indices (forced expiratory volume in 1 second, FEV1; FVC pred%, Carbon monoxide diffusion capacity, DLCO), and the patient's QoL, oncological, and fertility outcomes (lines 26 – 29, 91 – 96, 220 – 223).

Comment 2: In patients with bleomycin-induced acute lung injury, pulmonary function may not be measured due to severe symptoms. In addition, it may not be possible to measure pulmonary function in patients whose general condition has deteriorated due to the progression of MOGCT. Please show how to deal with such reduction bias.

Response: In patients who may not be able to receive pulmonary function tests due to bleomycin-induced acute lung injury, arterial blood gases (ABGs) and high-resolution pulmonary CT will be performed to indirectly examine the pulmonary function. Meanwhile, it is also an adverse event during this trial, we will compare whether there is a significantly different incidence of this event between the

experimental group and control group. A pulmonary function test will be performed after recovery or until those patients can complete this examination. For patients whose general condition has deteriorated due to MOGCT progression, if such patients had been enrolled in this trial, they would have recorded baseline and dynamic pulmonary function tests until they could not finish the pulmonary test. Therefore, we will also compare the existing data to determine whether nintedanib can alleviate bleomycin-induced pulmonary injury. Furthermore, if they have not already finished the complete treatment for MOGCT and cannot continuously receive bleomycin-based chemotherapy, these adverse events will be recorded as positive events during the trial, and they will be dropped out from the trial. Additional enrollment of MOGCT patients will be an alternative option to solve this issue (lines 173-184).

Comment 3: Please explain the reason why time-to-event with the onset of lung injury as an event is not endpoint.

Response: The main objective of our trial is to evaluate the protective effectiveness of nintedanib in pulmonary fibrosis induced by bleomycin, not the time to onset of lung injury in treating patients. Therefore, it is not the endpoint in this study. However, it could be an adverse effect that would be documented during the trial (lines 199 – 200).

Comment 4: Having chronic obstructive pulmonary disease or pulmonary hypertension influence pulmonary function. Patients with this disease were not included in the exclusion criteria.

Response: Thanks for your comments, we have added them as exclusion criteria in the revised manuscript (lines 134 -135).

Response to reviewer 2#: Thank you so much for your comments, the following are our responses:

Comment 1: The first one is about the age of the participants, which is 14-35y. Most of the similar trails set the age no older than 40 years old. As you mentioned in the limitations, it needs more time to enroll enough participants due to its rarity. I recommend extending the age of the included subjects to 40.

Response: Thanks for your suggestion, we have revised the included age as 14- 40 years in revision (line 121).

Comment 2: As I know that oxygen therapy, smoking, and chest irradiation are thought to synergize with bleomycin, I suggest the protocol point out that.

Response: We have pointed out the related potential factors that may have synergized effects with bleomycin (lines 194-196).

Comment 3: During the chemotherapy cycles, the participants may get pneumonitis and should stop the bleomycin. They can't finish the 3-4 cycles as the protocol mentioned, so how to deal with that?

Response: They would be removed from the trial in case of this circumstance; however, these adverse events would remain to be recorded and reported. Additional enrollment of MOGCT patients will be conducted if necessary (lines 173-184).

Comment 4: As I know, the sample size calculation is due to the primary outcomes of the study. Since it has several primary outcomes, the calculation of each one should be done to get the largest sample size. The protocol only mentioned the FVC and DLCO, and they were calculated at the same rate. I read the references and did not find the rate for the DLCO. I suggest consulting statisticians for this part, or I would propose only one primary outcome for the study.

Response: Thanks for your comments, we have modified the primary outcome as the decline of forced vital capacity (FVC). The decline of other pulmonary function indices (forced expiratory volume in 1 second, FEV1; FVC pred%, Carbon monoxide diffusion capacity, DLCO) will be recorded as the secondary outcomes of our study. Therefore, the related data (including changes of FEV1, FVC

pred%, and DLCO) will also be evaluated and obtained in MOGCT patients who received bleomycin after the accomplishment of this trial (lines 26 – 29, 91 – 96, 220 – 223).

All in all, we tried our best to improve the manuscript and made some changes to the revised manuscript. We believe these changes will make the manuscript more structured and consistent.

VERSION 2 – REVIEW

REVIEWER	Okamoto, Masaki Kurume University School of Medicine, Division of Respiriology, Neurology and Rheumatology, Department of Medicine
REVIEW RETURNED	28-Sep-2023

GENERAL COMMENTS	This paper had been revised correctly.
--

REVIEWER	Zhang, Wenxi Third Military Medical University Southwest Hospital, gynecology and obstetrics
REVIEW RETURNED	18-Oct-2023

GENERAL COMMENTS	Thank you! It's truly gratifying to see that most of the recommendations have been accepted. However, I still have a few suggestions below:  1.the "Recruitment of patients and data collection" section, specifically in Lines 179-183, the manuscript mentions that if patients have not completed the entire MOGCT treatment and cannot continue to receive bleomycin-based chemotherapy, any adverse events will be recorded as positive events during the trial, and these patients will be dropped from the study. In my opinion, it might be inappropriate to exclude participants due to adverse events after randomization, as the trial is intended to evaluate the drug's safety. Enrolling additional MOGCT patients might introduce bias into the trial. Given this scenario, I would recommend considering an increase in the estimated dropout rate and expanding the sample size. 2.Upon reevaluating the manuscript, I noticed that there is no "Harms" section, which I believe is crucial for a drug and chemotherapy trial. Especially, the protocol mentioned there is limited evidence about it. Please consider adding this section in accordance with the SPIRIT Checklist. 3. The other limitation of this study is that it's a single center study. Please address that.
--

VERSION 2 – AUTHOR RESPONSE

Response to reviewer 1#: Thank you for reviewing our manuscript.

Response to reviewer 2#: Thank you so much for your comments, the following are our responses:

Comment 1: the "Recruitment of patients and data collection" section, specifically in Lines 179-183, the manuscript mentions that if patients have not completed the entire MOGCT treatment and cannot

continue to receive bleomycin-based chemotherapy, any adverse events will be recorded as positive events during the trial, and these patients will be dropped from the study. In my opinion, it might be inappropriate to exclude participants due to adverse events after randomization, as the trial is intended to evaluate the drug's safety. Enrolling additional MOGCT patients might introduce bias into the trial. Given this scenario, I would recommend considering an increase in the estimated dropout rate and expanding the sample size.

Response: Thanks for your comment. Indeed, it's more appropriate as your suggestion since we also want to evaluate the safety of nintedanib in MOGCT. According to the current data, it is estimated that 8% of patients receiving bleomycin will develop varied degrees of pulmonary fibrosis, we presume that no more than 10% of enrolled MOGCT patients cannot continue to receive bleomycin-based chemotherapy. Namely, we will recruit additional 10% patients (12 patients) except for the previous 116 MOGCT patients to better conduct this trial. Therefore, we modify the final sample size as 128 patients in revision (line 241 – 244). The related expression has also been modified (line 22, 182).

Comment 2: Upon reevaluating the manuscript, I noticed that there is no "Harms" section, which I believe is crucial for a drug and chemotherapy trial. Especially, the protocol mentioned there is limited evidence about it. Please consider adding this section in accordance with the SPIRIT Checklist.

Response: The safety of nintedanib in treating idiopathic pulmonary fibrosis has been confirmed in previous clinical trials. The AEs are usually mild and tolerable. The most common AEs of nintedanib therapy are as follows (incidence $\geq 5\%$): diarrhea, anorexia, sickness, vomiting, abdominal pain, headache, elevated liver enzyme, weight loss, or hypertension. Other rare AEs reported in previous research include hemorrhage, gastrointestinal perforation, elevated bilirubin levels, and cardiac toxicity. However, currently, there has been limited data on the safety of nintedanib in MOGCT patients receiving bleomycin-based chemotherapy. Therefore, unpredicted AEs may be found during this trial, and all the AEs observed during the study period will be treated according the clinical practice. Especially, for MOGCT patients receiving bleomycin-based chemotherapy, the benefit of prevention of bleomycin-induced pulmonary toxicities would be greater than the risk and the harms of nintedanib itself (line 281 – 291).

Comment 3: The other limitation of this study is that it's a single center study. Please address that.

Response: As your suggestion, we add it as the major limitation in the discussion section (line 47, 331 – 333).

All in all, we tried our best to improve the manuscript and made some changes to the revised manuscript. We believe these changes will make the manuscript more structured and consistent.

Once again, we would like to express our great appreciation to you and the reviewers for your comments on our paper. Looking forward to hearing from you.